# Response of Seed Germination and Seedling Growth of Six Desert Shrubs to Different Moisture Levels under Greenhouse Conditions

**DOI:** 10.3390/biology13090747

**Published:** 2024-09-23

**Authors:** Yonghong Luo, Hui Yang, Xingfu Yan, Yongrui Ma, Shuhua Wei, Jiazhi Wang, Ziyu Cao, Zhong Zuo, Chunhui Yang, Jiming Cheng

**Affiliations:** 1College of Biological Science and Engineering, North Minzu University, Yinchuan 750021, China; 21915013@mail.imu.edu.cn (Y.L.); 20217525@nmu.edu.cn (H.Y.); 13198504489@163.com (Y.M.); czy200218@163.com (Z.C.); 2School of Ecology and Environment, Inner Mongolia University, Hohhot 010021, China; 3Institute of Forestry and Grassland Ecology, Ningxia Academy of Agricultural and Forestry Sciences, Yinchuan 750002, China; nxzuozhong@163.com; 4Ningxia Key Laboratory of Sand Control and Soil and Water Conservation, Yinchuan 750002, China; 5Key Laboratory of Ecological Protection of Agro-Pastoral Ecotones in the Yellow River Basin, National Ethnic Affairs Commission of the People’s Republic of China, Yinchuan 750004, China; 6Ningxia Academy of Agriculture and Forestry Sciences, Plant Protection Institute, Yinchuan 750002, China; weishuhua666@163.com; 7Chengde Meteorological Disaster Prevention Center of Hebei Province, Chengde 067000, China; george0602@126.com; 8School of Literature and Communication, China Three Gorges University, Yichang 443002, China; yangch6882@163.com; 9School of Life Sciences, Central China Normal University, Wuhan 430079, China

**Keywords:** *Caragana korshinskii*, *Calligonum mongolicu*, desert steppe, functional traits, grassland restoration, soil moisture, *Zygophyllum xanthoxylum*

## Abstract

**Simple Summary:**

The effects of different moisture contents on seed germination and seedling growth of six desert shrubs were determined in a greenhouse. It was found that 5% and 10% limited seed germination and seedling growth, while 15–20% resulted in the highest seed germination and biomass accumulation. In addition, we found that *Caragana korshinskii* had the strongest survivability, while *Nitraria sibirica* had the lowest viable survivability. Our results emphasise that cultivation of *Caragana korshinskii*, which is more drought tolerant, is preferred in desert areas for the recovery of vegetation in the region.

**Abstract:**

Moisture is the most important environmental factor limiting seed regeneration of shrubs in desert areas. Therefore, understanding the effects of moisture changes on seed germination, morphological and physiological traits of shrubs is essential for vegetation restoration in desert areas. In March to June 2023, in a greenhouse using the potting method, we tested the effects of soil moisture changes (5%, 10%, 15%, 20% and 25%) on seed germination and seedling growth of six desert shrubs (*Zygophyllum xanthoxylum*, *Nitraria sibirica*, *Calligonum mongolicum*, *Corethrodendron scoparium*, *Caragana korshinskii*, and *Corethrodendron fruticosu*). Results showed that (1) seed germination percent and vigor index were significantly higher at 15 and 20% soil moisture content than at 5 and 10%; (2) shoot length, primary root length, specific leaf area and biomass of seedlings were significantly higher in the 15% and 20% soil moisture content treatments than in the 5% and 10% treatments; (3) superoxide dismutase activity (SOD) and soluble protein content (SP) decreased with decreasing soil water content, while peroxidase activity (POD) and catalase activity (CAT) showed a decreasing and then increasing trend with increasing soil water content; (4) the six seeds and seedling of shrubs were ranked in order of their survivability in response to changes in soil moisture: *Caragana korshinskii* > *Zygophyllum xanthoxylum* > *Calligonum mongolicum* > *Corethrodendron scoparium* > *Corethrodendron fruticosu* > *Nitraria sibirica*. Our study shows that shrub seedlings respond to water changes by regulating morphological and physiological traits together. More importantly, we found that *C. korshinskii*, *Z. xanthoxylum* and *C. mongolicum* were more survivable when coping with water deficit or extreme precipitation. The results of the study may provide a reference for the selection and cultivation of similar shrubs in desert areas under frequent extreme droughts in the future.

## 1. Introduction

Vegetation restoration is a crucial step in the process of desert ecological restoration, which can effectively curb the continuous expansion of desertification [1]. Seed germination and seedling growth are bottlenecks in the process of vegetation regeneration [2,3], directly determining the establishment of vegetation community structure and the direction and rate of self-renewal of dominant established species in desert areas [3]. Many plants have dormancy mechanisms that prevent germination in response to external adverse environmental factors until conditions are favourable for seed germination and seedling survival and, therefore, germination characteristics can reflect the environmental conditions under which a species can successfully establish itself [4]. Soil moisture is one of the key environmental factors affecting plant growth and development, especially for desert plants, which are most sensitive to changes in soil moisture during seed germination and early seedling development [4]. In addition, it has been shown that the tolerance of different plants to drought stress was significantly different [5]. Consequently, studying the growth and development characteristics of different sand-fixing plants under soil moisture changes has been one of the hot issues in the field of ecology [6,7,8].

Plant functional traits are morphological, chemical and physical traits that play an important role in regulating plant survival, growth and reproduction [9,10]. Plant height affects the ability of plants to capture light [11]. Tall plants usually have high light competition ability, which affects their growth rate and biomass production, which makes plant height one of the most important functional traits of plants [12]. As the main site of plant photosynthesis, leaf area, specific leaf area, and other traits strongly affect the photosynthetic capacity of plants [13,14]. Leaves not only directly affect plant interception of light resources, but also jointly regulate plant response to drought stress with leaf morphology traits, such as leaf thickness and specific leaf area [15]. Plants grown in arid areas usually have small leaves and, as drought intensifies, the leaf area decreases, resulting in a decrease in specific leaf area [16]. This can effectively reduce water loss in the plant body and improve its water use efficiency [17]. The root system, as an organ for plants to absorb nutrients and water, also undergoes changes in response to adverse external environmental changes [18]. For example, in arid environments, plants experience an increase in root length, a decrease in root diameter, and an increase in specific root length, thereby increasing root water uptake and improving water use efficiency [19,20]. Additionally, plants can also respond to environmental changes by adjusting their membrane system, antioxidant enzyme system, osmotic regulation system, and other aspects to minimize the negative impact of environmental changes on plants [21,22].

*Zygophyllum xanthoxylum*, *Nitraria sibirica*, *Calligonum mongolicum*, *Corethrodendron scoparium*, *Caragana korshinskii* and *Corethrodendron fruticosu* are the main afforestation shrub species for desertification control in Northwest China, Kazakhstan, and Mongolia [23,24,25,26,27,28]. They are mainly distributed in the fixed and semi-fixed sandy land in arid and semi-arid areas. Under long-term extreme drought conditions, desert plant seeds have developed complex physiological and ecological feedback mechanisms to resist soil water stress, but the specific mechanisms may vary among species [29]. For example, the minimum soil moisture content for seed germination of *Haloxylon ammodendron* and *Ammopiptanthus mongolicus* is 8% [30,31], while the minimum soil moisture content for seed germination of *Atriplex patens* is 5% [31], and the minimum soil moisture content for seed germination of *Tetraena mongolica* is 10% [31]. On the contrary, when the soil moisture content exceeds 12.5%, there is a negative correlation between the germination rate of *Ammopiptanthus mongol* and soil moisture content [32]. Similarly, when the moisture content exceeds 19.4–20.5%, it inhibits the germination of *Artemisia sphaerocephala* seeds [33]. Previous studies have found that these species can bear a large number of seeds in wild desert habitats, but there are few seedlings of these six shrubs, indicating that their seed germination and seedling growth are severely limited by drought stress. Therefore, studying the relationship between seed germination and seedling growth and physiological characteristics of these six shrubs and their response to soil moisture can help to reveal their population renewal strategies in desert environments.

Due to global warming causing changes in rainfall patterns, models predict that extreme rainfall events will increase in the mid latitudes of the Northern Hemisphere, exacerbating drought and heavy rainfall events in desert ecosystems [34]. The changes in rainfall have a significant impact on seed germination and seedling growth, especially in desert grasslands with water deficiency. The aim of this study is to investigate the effects of different soil moisture contents on seed germination, seedling phenotype traits, and physiological characteristics. The following questions were raised: (1) How does soil water affect seed germination and seedling growth characteristics? (2) What are the drought adaptation mechanisms for seed germination and seedling growth of these six species?

## 2. Experimental Materials and Methods

### 2.1. Seed Collection and Storage

The experimental plant seeds were collected from the southern edge of the Maowusu Desert in Yanchi County, Ningxia Hui Autonomous Region, China (107°37′ E, 37°85′ N). Yanchi County’s climate is in a transitional zone from semi-arid to arid, belonging to a typical temperate continental climate. The annual average rainfall is 280 mm, and the annual evaporation is 2100 mm [35]. The vegetation is mainly composed of desertified grasslands and low, drought-tolerant, super-drought-tolerant shrubs and semi-shrubs. During August and September 2022, seeds of *Z. xanthoxylum*, *N. sibirica*, *C. mongolicum*, *C. scoparium*, *C. korshinskii*, and *C. fruticosu* were collected. During the collection, seeds were directly picked from four different directions on adult plants, with 20 plants selected for each species. After harvesting, the seeds were taken back to the laboratory. Artificially selected seeds had an intact appearance, with no mold, no insect feed, plump seeds, and uniform size. The seed mass and morphological characteristics of the six desert shrubs are shown in Table 1. After selection, the seeds were stored in breathable mesh bags at room temperature for use (manually removing the wings, flower stems, and pods of the *Eucommia ulmoides* seeds). Soak and disinfect the seeds in a 2.5% sodium hypochlorite solution for 10 min before use, rinse 7–8 times with sterile water, and use filter paper to absorb the surface moisture of the seeds before use.

### 2.2. Material Collection and Treatment

The experimental site was the greenhouse of the Biology Experimental Base of North Minzu University, the daytime and nighttime temperatures were (23 ± 5) °C, (16 ± 3) °C, respectively, and the light duration was about 8 h/day. In late March 2023, 180 plastic flowerpots (19.5 cm in height and 19.8 cm in outer diameter) were taken and evenly divided into 6 groups, with 30 pots in each group used for sowing and seedling cultivation of 6 shrub seeds. Divide the 30 flowerpots of each shrub into 5 groups and set them as soil moisture treatments of 5%, 10%, 15%, 20%, and 25%, respectively. Repeat each soil moisture treatment 6 times and sow 30 seeds in each pot. The number of seeds used = 30 seeds (each flowerpot) × 6 (replicates) × 5 (treatments) × 6 (species) = 5400 seeds. Collected sun-dried sandy soil from growing natural shrubs was placed in the flowerpots, and the weight of each flowerpot was 5.5 kg with the addition of sun-dried sand. The calculated soil moisture content was 5%, 10%, 15%, 20%, and 25%, and the weight was 5.77, 6.04, 6.31, 6.58, and 6.85 kg. We designed the following five soil moisture gradients with reference to the previous study of sand-fixing plant response to moisture in this region by Yang et al. [35]. After sowing, watering took place to the corresponding weight. The sowing depth of the seeds was 2 cm.

The standard germination test is if the radicle had emerged by at least 2 mm [36]. The number of germinated seeds was recorded every 3 days, and the embryo length (mm) above the surface of the germinated seeds was measured using a vernier caliper. After the emergence was complete, each pot should have retained 8 seedlings of similar size. During the experiment, the flowerpot was weighed every 3 days, and the water lost due to evaporation was replenished based on the mass of the flowerpot (including soil and seeds) at the beginning of the experiment (ignoring the mass changes during seed and seedling growth). Seedlings were harvested 90 days after treatment. Firstly, soak the sand thoroughly with water, then dig out all the seedlings, put them into self-sealing bags and bring them back to the laboratory. Rinse the soil from the roots and leaves of the seedlings with tap water, and then use dry filter paper to absorb the water droplets on the surface of the seedlings. Divide the harvested seedlings into two groups on average, one group used to measure the growth characteristics of the seedlings, and the other group to measure the physiological indicators of the plant leaves.

### 2.3. Determination of Seed Germination Parameters, Seedling Phenotype Traits, and Physiological Characteristics

Calculate germination parameters, such as germination percentage (GP) and vigor index (VI)m using the following formulas [37]:*GP* = *n* × 100%/*N*

where *n* and *N* are the numbers of germinated seeds and the total number of seeds used in the experiment [37].
*VI* = *GP* × [seedling root length (mm) + seedling stem length (mm)] 

The shoot length was determined using a straightedge (units: cm), and leaf area per plant was determined by American LI-3100 leaf area meter (unit: cm^2^). Seedlings were divided into roots, stems and leaves and then the plant tissues were placed in envelopes and inserted in a blast oven at a temperature of 65 °C, dried to a constant weight to determine total dry mass (TDM), root-shoot ratio (RSR), specific leaf area (SLA), specific shoot length (SSL) and specific root length (SRL); all the data were retained in two decimal places. The formulae were calculated as follows:

TDM = Root dry mass + stem dry mass + leaf dry mass [2]

RSR = (Root dry mass)/(stem and leaf dry mass) [2]

SLA = (Leaf area)/(leaf dry mass (cm^2^·g^−1^)) [2]

SSL = (shoot length)/(shoot dry mass (cm·g^−1^)) [2]

SRL = (Root length)/(root dry mass (cm·g^−1^)) [2]

After cutting and mixing the leaves of the seedlings, peroxidase activity (POD), catalase activity (CAT), superoxide dismutase activity (SOD), malondialdehyde content (MDA), soluble protein content (SP), and proline content (Pro) were measured using the guaiacol method [38], ultraviolet absorption method [38], nitrogen blue tetrazolium photoreduction method [38], thio-barbituric acid method [38], Coomassie Brilliant Blue G-250 method [38] and indole-3-acetic acid staining method [38], respectively.

### 2.4. Data Analysis

Seed germination parameters, seedling growth and physiological parameters among different species under the same soil moisture content were analysed using One-way ANOVA. One-way ANOVA was also used to test seed germination parameters, seedling growth and physiological parameters among different soil moisture contents. Calculation of the survivability of six desert shrubs to moisture changes: firstly, we standardised the data (including germination parameters, seedling growth parameters and physiological parameters). Secondly, inter-sample similarity analysis (decorana) was performed to understand the similarities and differences among samples, which would make subsequent rda analyses more reliable. Finally, we viewed each of the seed germination parameters, seedling growth parameters, and physiological parameters as having a consistent magnitude of effect on shrub survivability and performed a redundancy analysis (RDA) using the vegan package in R. Two different permutation tests were performed to assess the ability of the independent variables to influence each of the dependent variables, and 999 permutation tests were performed on the RDA model using the permutest function to assess the extent to which the overall independent variables explained each of the dependent variables. The envfit function was then used to perform 9999 permutation tests for each parameter factor to assess the amount of explanation of the dependent variable by each independent variable. All statistical analyses were performed in R (version 4.3.1) software.

## 3. Results and Analyses

### 3.1. Seed Germination Response to Soil Moisture

Seed germination percentage and vigour index showed an increasing and then decreasing trend with increasing soil moisture, and the highest germination percent and vigour index of plant seeds were observed at soil moisture content of 15–20%. The germination percent of *Z.xanthoxylum* seeds was higher than that of the other five plants (the average germination rates of *Z. xanthoxylum*, *N. sibirica*, *C. mongolicum*, *C. Scorpium*, *Caragana korshinskii*, and *C. fruticosu* seeds were 44.5%, 33.99%, 24.61%, 24.67%, 24.67%, and 23.67%, respectively). The vigour index of *Z. xanthoxylum* seeds was significantly higher than that of the other five plants under different soil moisture content treatments (Figure 1A,B).

### 3.2. Seedling Traits and Biomass Response to Soil Moisture

Shoot length, primary root length, and leaf area per plant increased and then decreased with increasing soil moisture content, and shoot length was highest at 15% and 20% soil moisture content (Figure 2A–C). Root-shoot ratio of plants was significantly higher at 5% than that of seedlings under the other four moisture treatments (Figure 2D). Specific leaf area of plants tended to increase and then decrease with increasing soil water content, with a minimum specific leaf area at 5% soil moisture content (Figure 2E). Seedling specific root length and specific shoot length were significantly higher at 5% and 10% soil moisture content than at the other three moisture treatments (Figure 2F,G). Seedling biomass tended to increase and then decrease with increasing soil moisture content. Seedling biomass was higher at 15% and 20% soil moisture content than at the other three moisture treatments. The biomass of *Z. xanthoxylum* seedlings was significantly higher than the biomass of seedlings of the other five plant species (Figure 2H).

### 3.3. Response of Physiological Characteristics of Seedlings to Soil Moisture

SOD concentration and proline content of seedlings decreased with increasing soil water content (Figure 3A,E). POD and CAT showed a decreasing and then increasing trend with increasing soil water content (Figure 3B,C). MDA and protein did not change significantly between water contents; MDA and protein contents of *C. korshinskii* were significantly greater than those of the other five species (Figure 3D,F).

The interpretation of *C. korshinskii*, *Z. xanthoxylum*, *C. mongolicum*, *C. scoparium*, *C. fruticosu* and *N. sibirica* for survivability was 64.48%, 45.30%, 26.17%, 7.67%, 1.18% and 0.14%, respectively (Figure 4). *C. korshinskii*, *Z. xanthoxylum* and *C. mongolicum* were significantly explained by survivability (*p* < 0.05), but *C. scoparium*, *C. fruticosu* and *N. sibirica* were not significantly explained by survivability (Figure 4).

## 4. Discussion

### 4.1. Seed Germination Response to Moisture

Water supply to the seed germination stage has a direct impact on plant establishment, growth and development, and is a key stage in the success of seeding afforestation in sandy areas. Studies have shown that either too low or too high soil moisture can lead to difficulties in seed germination [39,40]. This study found that seed germination percentage and vigour index of all six shrub plants showed an increasing and then decreasing trend with increasing soil moisture content and were maximum at 15–20%. Desert plants have evolved over time to produce a variety of adaptive responses, including seed germination methods. The seeds of plants are resistant to a variety of unfavourable environmental factors and, once transformed from seed to seedling, they become sensitive to the response of the external environment [41]. At 5% soil moisture content, seed germination of all six shrubs was less than 30%. Similarly, Lu et al. [42] found that drought stress significantly reduced germination rates of four desert plants (*Ixiolirion tataricum*, *Nepeta micrantha*, *Lepidium apetalum* and *Plantago minuta*) selected from Xinjiang. This can be explained by the following four aspects. Firstly, the low soil moisture prevents the seed coat from softening, limiting the entry of oxygen and preventing the seeds from breathing and metabolizing. Secondly, lack of moisture does not change the cellular protoplasmic state, and effective moisture promotes changes in the protoplasm in the seed embryo from an inactive gel state to an active sol-gel state. Thirdly, water is the medium of decomposition of the stored material in the seed; without water, all kinds of organic nutrients cannot be decomposed into simple and easy small molecules for the seed to absorb [43]. Eventually, drought stress increases the synthesis of abscisic acid in plants and inhibits the germination of plant seeds [44]. On the contrary, the lack of oxygen caused by high soil water content and the products of anaerobic respiration have a toxic effect on seeds and thus also inhibit seed germination [45]. In addition, this study found differences in seed germination percentage and vigour index among the species. For example, *Z. xanthoxylum* seed germination was significantly higher than the other five species at 5–10% moisture content, suggesting that *Z. xanthoxylum* seeds are more advantageous in transforming into seedlings when moisture is lacking. Maximum germination of the seeds of the plants was achieved at 15–20% moisture content, but only *Z. xanthoxylum* and *N. sibirica* had seed germination rates of 50% for both. This is probably because seed germination is not only affected by environmental factors, such as moisture, but could also be related to dormant factors, such as seed size, morphology, seed coat thickness, nutrient reserves, and endogenous hormone content [46]. Some of the seeds remain dormant and enter the soil seed bank in order to cope with the mass mortality of seedlings due to the lack of effective rainfall after concentrated seed germination [47]. Hence, a conservative germination strategy is used to ensure the continuation of the population in desert ecosystems by germinating at different times [48].

### 4.2. Seedling Growth Response to Moisture

Moisture is an important environmental factor that affects plant growth. Lack of moisture reduces seedling height and root length [49], decreases leaf area and specific leaf area [34], and hinders dry matter accumulation [50]. We found that, at lower water content (5% and 10%), seedlings had lower shoot length, primary root length, leaf area per plant, and specific leaf area than in the other three moisture treatments. This is because of reducing shoot length, leaf area and specific leaf area, via which plants can reduce their ability to capture light and evaporate water, and increase water use efficiency [11], thereby improving plant survival under drought stress [34]. In desert steppe, Li et al. [50] found that drought significantly reduced plant height and aboveground biomass accumulation through four consecutive years of controlled experiments. Similarly, Ye et al. [51] found that drought stress significantly reduced plant height and leaf area of *Stipa breviflora*. At 25% soil moisture content, seedling growth was primary and specific root lengths of plants were significantly higher than those of the other slowed down, because more moisture reduces the permeability of the soil, leading to root hypoxia and reduced growth rates [52]. Additionally, this study found that the root-shoot ratio of seedlings was significantly higher than in the other four moisture treatments at 5% soil moisture content and the three moisture treatments at soil moisture contents of 5% and 10%. The allometric partitioning theory suggests that, when a plant is constrained by a resource, it allocates more biomass to the organ responsible for capturing that resource [53]. Under drought stress, plants reduce the area of above-ground organs intercepting solar radiation in order to avoid excessive water loss, and seedlings preferentially allocate more photosynthetic products to the root system to improve the plant’s ability to access water resources from deeper soil layers [54]. Increasing specific root length is effective in increasing the water use efficiency of plants to access limited environmental resources and maintain optimal plant survival [55].

### 4.3. Seedling Physiological Characteristics in Response to Moisture

Under drought stress, leaf stomatal closure leads to carbon dioxide diffusion, electron transfer obstruction and physiological and biochemical disorders, generating excessive reactive oxygen species (ROS) resulting in oxidative stress [56], causing membrane lipid peroxidation products and increased plasma membrane permeability, and damaging the membrane system [57].

Meanwhile, plant cells also activate the defensive enzymes SOD, POD and CAT to remove ROS and MDA in order to maintain the stability of the cell membrane and the normal functioning of the photosynthetic apparatus [58]. In this study, it was found that 5% soil moisture content treatment increased SOD activity, POD and CAT in seedlings, which was attributed to the mobilisation of enzymes, such as SOD, POD and CAT, by the plant cells in a synergistic manner to scavenge ROS in response to drought stress [58]. With the increase in soil water content, SOD showed a decreasing trend, while CAT and POD decreased and then increased, probably because the latter two played a more important regulatory role when the soil water content was higher. Additionally, this study found that the proline content at lower soil water content (5–10%) was higher than the proline content at higher water content treatments (15–25%). By accumulating proline to reduce cellular water potential when plants are subjected to drought stress, this prevents cell membrane dissociation, enhances cellular water retention and stabilises cellular structure [59]. Simultaneous scavenging of oxygen radicals provides energy for plant seedling growth after drought stress [60]. Seedling MDA and soluble protein contents did not change significantly among treatments but differed among species, which may be related to differences in the strategies of different plants to cope with drought stress.

### 4.4. Survivability of Six Desert Shrubs in Response to Changes in Moisture

Comprehensive analysis of seed germination parameters, growth characteristics and physiological properties of seedlings of six desert plants using RDA revealed that *C. korshinskii* had the highest survivability followed by *Z. xanthoxylum* and *C. mongolicum*, and the degree of explanation of the survivability of all the three reached the level of significance, while *C. scoparium*, *C. fruticosu* and *N. sibirica* did not reach the level of significance in explaining the survivability. This was consistent with the distribution of shrub vegetation in our study area, with *C. korshinskii* having the highest distribution area, accounting for about 80% of the scrub area.

## 5. Conclusions

We tested the response of seed germination and seedling growth of six shrubs to moisture in a greenhouse and found that the optimum moisture for seed germination was 15–20%. In addition, we found that 15–20% was also the most favourable for seedling growth by observing the phenotypic traits and physiological characteristics of shrub seedlings. More importantly, we comprehensively analysed the seed germination parameters, seedling growth parameters, and physiological parameters of six shrubs using RDA, and found that the order of survivability was as follows: *Caragana korshinskii* > *Zygophyllum xanthoxylum* > *Calligonum mongolicum* > *Corethrodendron scoparium* > *Corethrodendron fruticosu* > *Nitraria sibirica*. These results are informative for the selection and breeding of shrubs in similar desert areas. However, our study only focused on moisture and, in later studies, we will focus on the effects of other environmental factors (e.g., temperature and salinity) [37] and plant-animal interactions [60] on the seedling growth of seed germination of these shrubs.

## Figures and Tables

**Figure 1 biology-13-00747-f001:**
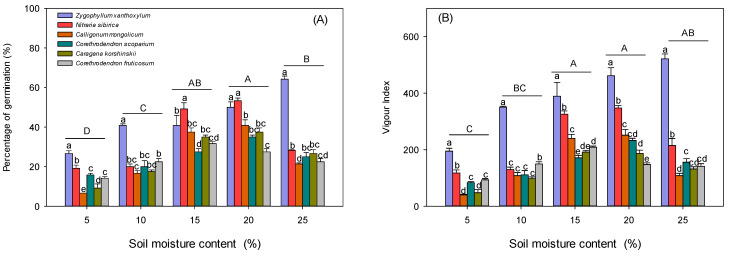
Effect of soil moisture content on germination percent and vigour index of different desert plants. (**A**) Percentage of germination, (**B**) vigour index. Different lower-case letters indicate significant differences in seed germination parameters between species at the same soil moisture content, and different upper-case letters indicate significant differences in seed germination parameters between moisture contents (*p* < 0.05).

**Figure 2 biology-13-00747-f002:**
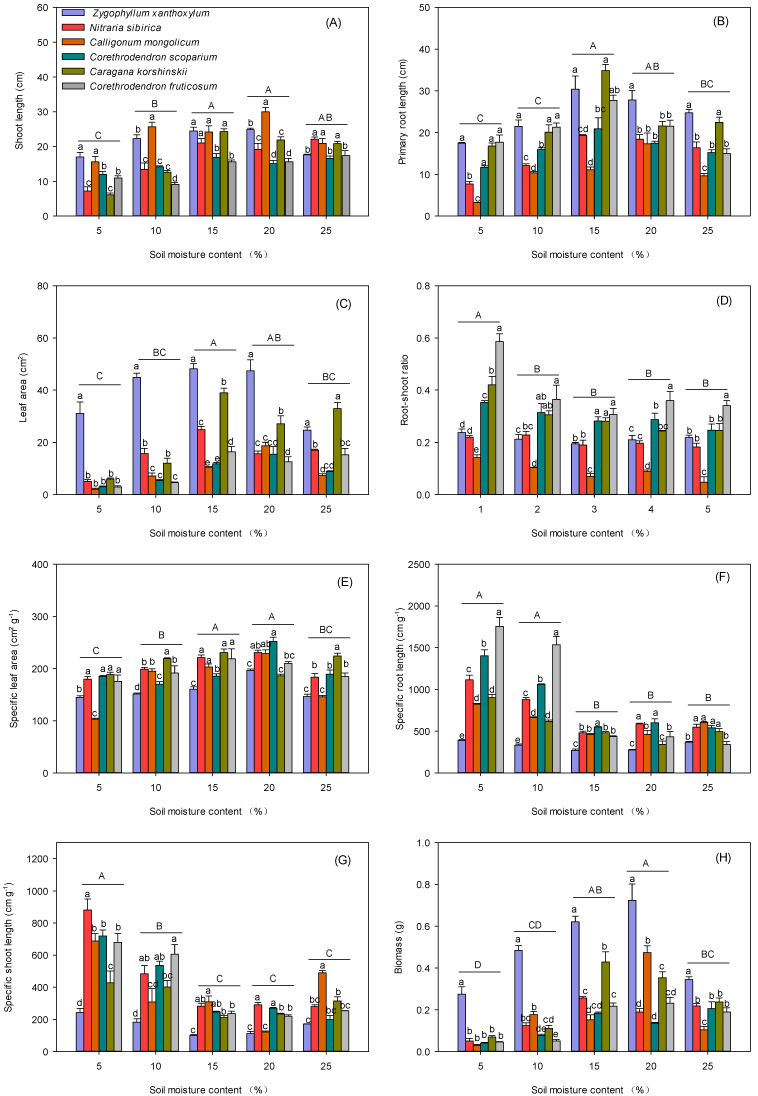
Effect of soil moisture content on phenotypic traits and biomass of different desert scrub plants. (**A**) Shoot length, (**B**) primary root length, (**C**) leaf area, (**D**) root-shoot ratio, (**E**) Specific leaf area, (**F**) specific root length, (**G**) specific shoot length, (**H**) biomass. Different lower-case letters indicate significant differences in seedling growth parameters between species at the same soil moisture content, and different upper-case letters indicate significant differences in seedling growth parameters between moisture contents (*p* < 0.05).

**Figure 3 biology-13-00747-f003:**
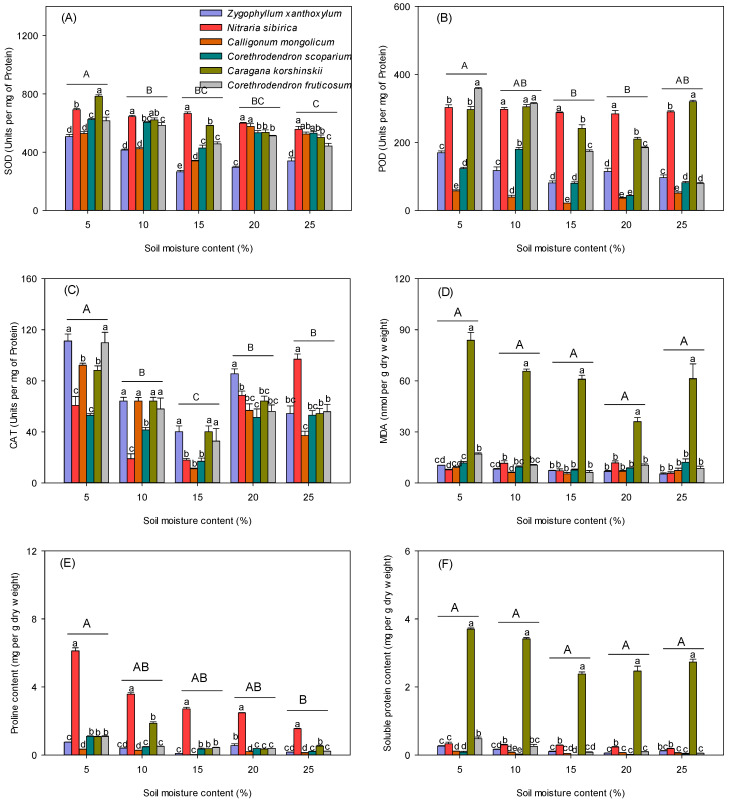
Effect of soil moisture content on physiological characteristics of seedlings of different desert scrub plants. (**A**) SOD concentration, (**B**) POD concentration, (**C**) CAT content, (**D**) MDA content, (**E**) proline content, (**F**) soluble protein content. Different lower-case letters indicate significant differences in seedling physiological parameters between species at the same soil moisture content, and different upper-case letters indicate significant differences in seedling physiological parameters between moisture contents (*p* < 0.05).

**Figure 4 biology-13-00747-f004:**
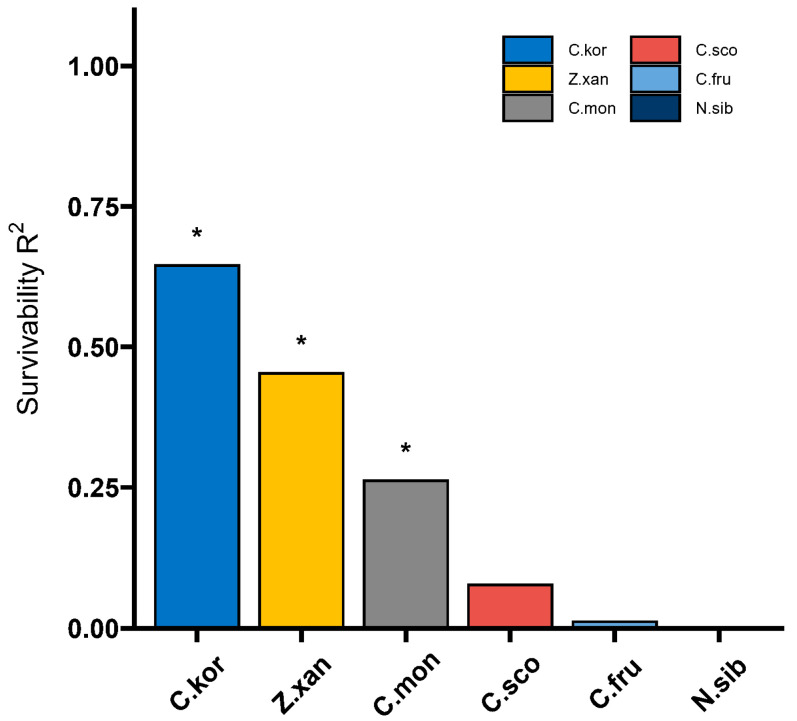
Explanation of soil moisture response to survivability in six desert shrubs. * indicates that the interpretation of independent variable has a significant effect on the dependent variable (*p* < 0.05). C.kor (*C. korshinskii*), Z.xan (*Z. xanthoxylum*), C.mon (*C. mongolicum*), C.sco (*C. scoparium*), C.fru (*C. fruticosu*), N.sib (*N. sibirica*).

**Table 1 biology-13-00747-t001:** Seed mass and morphology of six desert shrubs.

Species	*Zygophyllum xanthoxylum*	*Nitraria sibirica*	*Calligonum mongolicum*	*Corethrodendron scoparium*	*Caragana korshinskii*	*Corethrodendron fruticosu*
Families	Zygophyllaceae	Nitrariaceae	Polygonaceae	Fabaceae	Fabaceae	Fabaceae
Seed mass (mg)	21.53 ± 0.67	10.8 ± 0.69	94.06 ± 3.3	16.53 ± 0.61	40.33 ± 0.61	8.93 ± 0.35
Seed morphology	Crescent shape	Egg shape	Ellipse shape	Round kidney shape	Rectangular circle shape	Flat circular shape

## Data Availability

Dataset available upon request from the authors.

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
