# Peer review of "Response of Seed Germination and Seedling Growth of Six Desert Shrubs to Different Moisture Levels under Greenhouse Conditions"

_biology, 2024, doi:10.3390/biology13090747_

Round 1
Reviewer 1 Report
Comments and Suggestions for Authors
All questions of and suggestions for authors are in PDF file.

English writing is poor and should be improved, especially in materials and methods.
Author Response
Question1 The title dose not well represent your work. As you performed your experiment under greenhouse conditions, it must be mentioned in the title and abstract. Thus, the new proposed title is “Response of seed germination and seedling growth of six desert scrub plants to soil moisture content under greenhouse conditions”
Response1:We fully accept the reviewer's suggestion and have revised the title“Response of seed germination and seedling growth of six desert scrub plants to soil moisture content under greenhouse conditions”.
Question2 Why do you use “scrub plants” instead of shrubs? Are there main differences between them? If yes, please explain in the text.
Response2:We've replaced scrub plants with shrubs.
Question3 Abstract is not clear and complete. The main reason of doing this work is not mentioned in abstract. Materials and methods are not explained in abstract and nobody can understand that this work has been performed under greenhouse conditions.
Response3:
Question4:What do you exactly mean by phenotypic traits? It seems that using morphological and physiological traits makes better sense.
Response4:We have replaced phenotypic traits with morphological and physiological traits.
Question5: Are these shrubs C3 plants?
Response5: Zygophyllum xanthoxylum belongs to the CAM (Crassulacean acid metabolism)plant. Calligonum mongolicum belongs to C4 plant. Nitraria sibirica,Corethrodendron scoparium,Caragana korshinskii and Corethrodendron fruticosu belongs to C3 plant.
Question6 :Please use one of these phrases: ‘Seed germination percent’ or ‘Percentage of seed germination’ (not seed germination percentage).
Response6: ‘Seed germination percent’ replaces ‘seed germination percentage’.
Question7: seeds and seedling of shrubs not shrubs because root system and plant vigor is completely different between seedlings and mature plants.
Response7: Thanks for the heads up, we have replaced shrub with seeds and seedling of shrubs.
Question8: what do you exactly mean by phenotypic traits? it seems that morphological and physiological traits make more sense.
Response8: We have replaced phenotypic traits with morphological and physiological traits.
Question 9: Replace scarcity with deficit.
Response9: Already modified.
Question 10: Replace the results showed with it has been shown.
Response10: Already modified.
Question 11: Scientific names should be italic.
Response11: Already modified.
Question 12: tree or shrub. Please be consistent in whole the text.
Response12: Already modified.
Question 13: Replace resistant with tolerant.
Response13: Already modified.
Question 14:Replace will be collected with were collected.
Response14: Already modified.
Question 15:Please use the past tense verbs to describe your work.
Response15: Already modified.
Question 16: Tense of verbs is not appropriate. Please change them.
Response16: Already modified.
Question 17: Please elaborate greenhouse climate conditions.
Response17: Already added
Question 18: Why did you use these levels of moisture content?
Response18: This is indeed a good question, and we designed the following five soil moisture gradients with reference to the previous study of sand-fixing plant response to moisture in this region by Yang et al.
Yang, Z.; Wang, L.;Zhang, X.; et al., Seed germination and seedling growth of typical sand-fixing plants in response to soil moisture. Arid Zone Research. 2023, 41,840-842.
Question 18: What kind of soil did you use? 100% sand? From where did you provide it? It was the soil of natural habitat of the shrubs?
Response18: Collection of sandy soil from shrubs growing in their natural habitat.
Question 19:”the weight is 5.5 kg”not a good and clear sentence.
Response19: we rewrote the sentence.” the weight of each flowerpot was 5.5 kg with the addition of sun-dried sand”
Question 20: Please talk about germination requirements. Do these seeds have and dormancy?
Response 20: The standard of germination test is the radicle had emerged by at least 2 mm (luo et al., 2022). These seeds have dormancy, soak and disinfect the seeds in a 2.5% sodium hypochlorite solution for 10 minutes before use.
Luo, Y.; Cheng, J.; Yan, X.; Zhang, J.; Zhang, J. Germination of seeds subjected to temperature and water availability: Implications for ecological restoration. Forests, 2022, 13, 1854.
Question 21: The number of germinated seeds was recorded every 3 days instead of the number of seed germination is recorded every 3 days,
Response 21: Already modified.
Question 22: What are the numbers at the end of each formula? Please provide at least one reference for each formula. Please explain components of the formula.
Response 22: We have removed the numbers at the end of the formula, added references, and explained the components of the formula.
Question 23: specific shoot length (SSL) and specific root length. Are these two traits common in seed science studies?
Response 23:
Question 24: Shoot height or length? please be consistent in whole the text. if shoots are vertical = shoot height if shoots are horizontal = shoot length
Response 24: We've changed the full text to shoot length.
Question 25: Please provide reference for each protocol.
Response 25: Reference added.
Question 26: It is not clear how you calculate survivability.
Response 26: We rewrote how to calculate survivability
Question 27: The results are actually presented in a qualitative way instead of quantitative presentation. The authors should show the differences among species by indicating the percentage or fold increase/decrease in means.
Response 27: We have added quantitative descriptions
Question 28: Figures have low quality. Please increase their resolution in final version.
Response 28: Thank you for your suggestion. We have increased the resolution of the figures
Question 29: It seems that 25% moisture in a sandy soil is not too high to adversely affect seed germination.
Response 29: the negative impact of 25% moisture content on seed germination is not too high,
Reviewer 2 Report
Comments and Suggestions for Authors
Manuscript by Yonghong Luo and colleagues examines seed germination of six desert shrubs at five soil moisture contents and examined seedling growth at different water regimes. Overall, the manuscript is interesting and falls within the scope of the journal. However, in order to improve its quality further, I would like to suggest following changes in the manuscript before its final acceptance for publication:
1. Title: Replace “water” with “different moisture levels”.
2. Abstract: Replace “(1) Results showed that” with “Results showed that (1)”.
3. Introduction (Second last paragraph’s third line): Most studied species are not ‘tree’ and therefore authors should replace it with any other suitable terminology such as “woody species” or “Shrub”. Also italicize species names here and throughout the manuscript.
4. Introduction (Third Last paragraph): This paragraph deals mainly with adaptive features of mature vegetative plants and has little/nothing about early life-history adaptations. Therefore authors should replace this text with description about drought/water-limitation adaptations during seed germination and early seedling stage.
5. Section 2.1 (First sentence): Add seed collection date (i.e. month with year). Also convert all sentences into past tense instead of future and present tenses throughout the methods and result sections.
6. Section 2.2: Please add ambient temperature and light duration during experiments.
7. Section 2.2: Instead of mentioning that the harvest was done at the end of June, please provide number days since the start of the treatment.
8. Section 2.3: Authors should provide brief description of extraction methods for enzymes and other biochemical assays. In addition, they should also cite references of the biochemical assays used.
9. Results and Fig 3: generally, enzyme activities are expressed as “Units per mg of Protein” and therefore authors should convert this data in this unit instead of expressing on per gram biomass. Protein should also be expressed as mg per g dry weight.
10. Fig 4: What is the basis/formula and rationale of using the parameter of Survivability? Authors should provide brief description with relevant reference on relevant occasion.
11. Section 4.1: Data showed that the Z. xanthoxylum seeds had higher germination than other species at 5-10% moisture. Authors should discuss what are some specific morphological or biochemical adaptations of the Z. xanthoxylum seeds that can (at least putatively) be responsible for the higher tolerance.
12. I suggest performing a correlation matrix analysis of the seedling growth and various biochemical data, so that it will be easier to point out which biochemical parameters are positively related to seedling growth and is there are a general pattern across species.
13. Language of the manuscript needs substantial improvement.
Comments on the Quality of English Language
Language of the manuscript needs improvement.
Author Response
Question1 Manuscript by Yonghong Luo and colleagues examines seed germination of six desert shrubs at five soil moisture contents and examined seedling growth at different water regimes. Overall, the manuscript is interesting and falls within the scope of the journal. However, in order to improve its quality further, I would like to suggest following changes in the manuscript before its final acceptance for publication:
Response 1 We are very grateful to the reviewers for recognising our research work, and we have revised the manuscript individually according to the reviewer's suggestions1.
Question 2 Title: Replace “water” with “different moisture levels”.
Response 2 Response 1
Question3: Abstract: Replace “(1) Results showed that” with “Results showed that (1)”
Response 3: Already modified
Question4:. Introduction (Second last paragraph’s third line): Most studied species are not ‘tree’ and therefore authors should replace it with any other suitable terminology such as “woody species” or “Shrub”. Also italicize species names here and throughout the manuscript.
Response 4: Thanks for the heads up, already modified.
Question 5.Introduction (Third Last paragraph): This paragraph deals mainly with adaptive features of mature vegetative plants and has little/nothing about early life-history adaptations. Therefore authors should replace this text with description about drought/water-limitation adaptations during seed germination and early seedling stage.
Response 5: Thanks for the heads up, already modified..
Question 6.Section 2.1 (First sentence): Add seed collection date (i.e. month with year). Also convert all sentences into past tense instead of future and present tenses throughout the methods and result sections.
Response 6: Thank you for your suggestion, we have modified it.
Question 7.Section 2.2: Please add ambient temperature and light duration during experiments.
Response 7: Already added.
Question 8.Section 2.2: Instead of mentioning that the harvest was done at the end of June, please provide number days since the start of the treatment.
Response 8: Seedlings harvested at the end of June have been replaced with seedlings harvested 90 days after treatment.
Question 9.Section 2.3: Authors should provide brief description of extraction methods for enzymes and other biochemical assays. In addition, they should also cite references of the biochemical assays used.
Response 9: Enzyme extraction methods and other biochemical assays have been simplified and references have been added.
Question 10.Results and Fig 3: generally, enzyme activities are expressed as “Units per mg of Protein” and therefore authors should convert this data in this unit instead of expressing on per gram biomass. Protein should also be expressed as mg per g dry weight.
Response 10::Already modified
Question 11.Fig 4: What is the basis/formula and rationale of using the parameter of Survivability? Authors should provide brief description with relevant reference on relevant occasion.
Response 11:This is indeed a good question. The ability of seedlings of shrubs to settle depends on two stages: seed germination and seedling growth . Since drought affects both seed germination and seedling growth. We firstly will use the seed germination parameters, phenotypic traits parameters and physiological characteristics parameters of the indicators in the standardised treatment after, here each indicator on the survival of the shrubs survivability impact consistent, a comprehensive evaluation of the size of the survivability of the survival of each shrub...
Question 12. Section 4.1: Data showed that the Z. xanthoxylum seeds had higher germination than other species at 5-10% moisture. Authors should discuss what are some specific morphological or biochemical adaptations of the Z. xanthoxylum seeds that can (at least putatively) be responsible for the higher tolerance.
Response 12: Due to reviewer 1's suggestion that we add quantitative descriptions, we deleted the statement that Z. xanthoxylum seed 5-10% germination was higher than that of the other five species in the results, so we are de-emphasising this part in the discussion..
Question 13.I suggest performing a correlation matrix analysis of the seedling growth and various biochemical data, so that it will be easier to point out which biochemical parameters are positively related to seedling growth and is there are a general pattern across species.
Response 13 This is indeed good advice, however, we used RDA in the manuscript to analyse the viability of each species in an integrated manner, where phenotypic traits and physiological characteristics are included. So we did not perform further analyses of correlations between these parameters.
Question 14. .Language of the manuscript needs substantial improvement.
Response 14 We have embellished the language of the manuscript.
Reviewer 3 Report
Comments and Suggestions for Authors
The most significant comment: the size of the article needs to be increased by reworking the abstract and literature review. It is also necessary to add the Conclusions or Conclusion section, which is missing from the article. Therefore, it is difficult to understand the result of the work done and assess the prospects for using the results obtained.
The title of the article generally reflects its content.
Abstract
The abstract should be expanded:
1. It is necessary to indicate in which region and in what years the research was conducted.
2. Briefly explain the research methodology.
3. More clearly define the purpose of the research and its scientific novelty.
4. Expand the resulting part, add numerical results, show the difference in options.
5. The abstract ends with a very vague and unclear sentence: "The results of our study have some informative value for the selection and cultivation of desert grassland shrubs." This is not how to end an article that will be read by researchers all over the world. It is necessary to finish the abstract with a conclusion that will be useful to other scientists who are studying similar issues in other regions.
Correction of these comments will make the abstract more informative and useful to researchers.
Key words: should be expanded in accordance with the content, add a list of studied species.
1. Introduction
The authors provide an overview of scientific data on the studied problem, out of 53 sources - 31 (58.5%) for the last 2019-2024. Also included are works starting from 1978, which shows the depth of elaboration of scientific material.
There is no excessive self-citation.
The references are correct and correspond to the presented material.
Recommendation:
1. The literature review should be revised and significantly increased. The authors write a lot of general phrases that do not provide readers with information about research on the issues of dormancy and seed germination. For example: "Plant functional traits are morphological, chemical and physical traits that play an important role in regulating plant survival, growth and reproduction [9,10]. Plant height affects the ability of plants to capture light [11]." and others.
2. It is necessary to explain to readers why dormancy issues and seed germination mechanisms are important for arid regions. Why different moisture regimes need to be studied when assessing dormancy exit, etc.
3. It is necessary to add information about the importance of the studied crops for agriculture in the desert zone. Indicate which families these crops belong to. The authors very briefly spoke about the importance of these crops in arid regions.
4. It is necessary to show readers in which other countries the studied crops are widespread. This will increase interest in the work of scientists from other countries.
The purpose of the work: is not clearly stated. In the objective of the work, the authors indicated: "The aim of this study is to investigate the effects of different temperature and water content on seed germination and evaluate the growth characteristics of seeds under different water supply systems." However, the methodology and the result do not discuss the effect of temperature. It is necessary to correct the goal of the work.
I recommend ending the review with a more clear (precise) statement of the goal and definition of the research objectives. And further in the text of the work, focus the readers' attention on the results of achieving the stated goal and objectives.
Experimental Materials and Methods
2.1. Seed collection and storage
The section is written in sufficient detail. The methods are reproducible. However, it is not clear whether the authors developed the methods of selection and processing of seeds independently, or there are recommendations in accordance with which the following selections were made: "During the collection, seeds will be directly picked from four different directions on adult plants, with 20 plants selected for each species."? Why a sample of 20 plants, and not 10 or 30? If these are standard recommendations, then this should be said and a link provided. If this methodology was developed by the authors, then this should also be stated in the text
2.2. Material Handling and Collection
The section is written in sufficient detail. The methods are reproducible.
It is necessary to explain why these particular soil moisture levels were taken: "The calculated soil moisture content is 5%, 10%, 15%, 20%, and 25%...". If these are standard recommendations, then this should be stated and a link provided. If these water concentrations were chosen independently, then this choice should be explained.
2.3. Determination of seed germination parameters, seedling phenotype traits, and physiological characteristics
The section is written in sufficient detail. The methods are reproducible.
It is necessary to explain the "vigor index (VI)" indicator. This indicator is not known in all countries. It is advisable to explain the source of the formula (provide a link). Is this index generally accepted, proposed by a specific scientist, or proposed by the authors? This will clarify for readers whether this index really reflects the germination energy or is it about something else?
2.4. Data Analysis
The section is written in sufficient detail. The methods are reproducible. No comments.
3. Results and analyses
3.1. Seed germination response to soil moisture
The section is written in sufficient detail, the necessary explanations are given, the figures illustrate the studied processes. No comments.
3.2. Seedling traits and biomass response to soil moisture
The section is written in sufficient detail, the necessary explanations are given, the figures illustrate the studied processes. No comments.
3.3. Response of physiological characteristics of seedlings to soil moisture
The section is written in sufficient detail, the necessary explanations are given, the figures illustrate the studied processes. No comments.
Discussion
4.1. Seed germination response to moisture
4.2. Seedling growth response to moisture
4.3. Seedling physiological characteristics in response to moisture
4.4. Survivability of six desert shrubs in response to changes in moisture
All parts of the Discussion section contain an extended discussion of the results obtained. A detailed comparison of the results obtained by the authors with the literature data is given. There are no comments on this section.
The Conclusions section is missing!
I recommend showing in the conclusions that the goal that the authors set when planning the experiment has been achieved. And more precisely indicate how the results will be used in the future - for scientific and industrial purposes. Can the obtained results be applied in other regions of the country and the world with similar soil and climate conditions?
The authors should very clearly emphasize in their conclusions the significance of the conducted research for other countries!
After making corrections and adding the Inputs section, the article can be recommended for publication.
I wish the authors successful work!

Author Response
Reviewer 3
Question The article is devoted to a really important and interesting issue related to the features of germination and formation of seedlings in six species of desert shrubs against the background of different soil moisture.In the current conditions of the expansion of desertification zones in the world, loss of biodiversity against the background of climatic disturbances, scientific research on the study of seed science of wild resistant species is becoming important. The article is novel and original. The results are processed by statistical methods. Their reliability is beyond doubt. The relevance of the problem that the authors chose to study is beyond doubt. Unfortunately, I do not speak English and cannot assess the correctness of the presentation of the material of the article in this language. In this regard, the publication of the article is undoubtedly necessary. However, there are a number of significant comments that need to be taken into account and corrections made to the text of the article.
The most significant comment: the size of the article needs to be increased by reworking the abstract and literature review. It is also necessary to add the Conclusions or Conclusion section, which is missing from the article. Therefore, it is difficult to understand the result of the work done and assess the prospects for using the results obtained. Unfortunately, the text of the article that was sent to me for review does not have a numbered term - this made it difficult to prepare a review and explanations for the authors. If the authors or editors have any questions about my comments, please send them and I will provide the necessary explanations.
Response:We are very grateful to the reviewers for their recognition of this study, and we have carefully revised the manuscript in accordance with the reviewers' suggestions.
Abstract
The abstract should be expanded:
Question1 It is necessary to indicate in which region and in what years the research was conducted.
Response1:Experimental sites and times have been supplemented.
Question2. Briefly explain the research methodology.
Response2:research methodology have been added.
Question3. More clearly define the purpose of the research and its scientific novelty.
Response3: We clarified the purpose and its scientific novelty of the study.
Question4: Expand the resulting part, add numerical results, show the difference in options.
Response4: We added a description of the variability in the results.
Question5: The abstract ends with a very vague and unclear sentence: "The results of our study have some informative value for the selection and cultivation of desert grassland shrubs." This is not how to end an article that will be read by researchers all over the world. It is necessary to finish the abstract with a conclusion that will be useful to other scientists who are studying similar issues in other regions.
Response 5: "The results of our study have some informative value for the selection and cultivation of desert grassland shrubs." has been replaced with “The results of the study may provide a reference for the selection and cultivation of similar shrubs in desert areas under frequent extreme droughts in the future”.
Question6: Key words: should be expanded in accordance with the content, add a list of studied species.
Response 6: In keywords: added: Caragana korshinskii; Calligonum mongolicu; Zygophyllum xanthoxylum;
- Introduction
The authors provide an overview of scientific data on the studied problem, out of 53 sources
– 31 (58.5%) for the last 2019-2024. Also included are works starting from 1978, which shows the depth of elaboration of scientific material. There is no excessive self-citation. The references are correct and correspond to the presented material.
Response: We thank the reviewer for their commendation of our study.
1 Recommendation:
Question7 : The literature review should be revised and significantly increased. The authors write a lot of general phrases that do not provide readers with information about research on the issues of dormancy and seed germination. For example: "Plant functional traits are morphological, chemical and physical traits that play an important role in regulating plant survival, growth and reproduction [9,10]. Plant height affects the ability of plants to capture light [11]." and others.
Response: 7 Thank you for your suggestion. We have added content on the impact of drought on seed germination in the third paragraph.
Question 8. It is necessary to explain to readers why dormancy issues and seed germination mechanisms are important for arid regions. Why different moisture regimes need to be studied when assessing dormancy exit, etc.
Response: 8 Thank you for your suggestion. We have added the issue of seed dormancy and explored different water content research in the third paragraph.
Question 9. It is necessary to add information about the importance of the studied crops for agriculture in the desert zone. Indicate which families these crops belong to. The authors very briefly spoke about the importance of these crops in arid regions.
Response: 9 We have supplemented the families to which these species belong in the method and increased the importance of these species.
Question 10. It is necessary to show readers in which other countries the studied crops are widespread. This will increase interest in the work of scientists from other countries.
Response: 10 We have added relevant content”Zygophyllum xanthoxylum, Nitraria sibirica, Calligonum mongolicum, Corethrodendron scoparium, Caragana korshinskii and Corethrodendron fruticosu are the main afforestation shrub species for desertification control in Northwest China, Kazakhstan and Mongolia”
Question 11. The purpose of the work: is not clearly stated. In the objective of the work, the authors indicated: "The aim of this study is to investigate the effects of different temperature and water content on seed germination and evaluate the growth characteristics of seeds under different water supply systems." However, the methodology and the result do not discuss the effect of temperature. It is necessary to correct the goal of the work. I recommend ending the review with a more clear (precise) statement of the goal and definition of the research objectives. And further in the text of the work, focus the readers' attention on the results of achieving the stated goal and objectives.
Response: 11:Thanks to your suggestions, we have rewritten this section with clear research objectives. "Due to global warming causing changes in rainfall patterns, models predict that extreme rainfall events will increase in the mid latitudes of the Northern Hemisphere, exacerbating drought and heavy rainfall events in desert ecosystems . The changes in rainfall have a significant impact on seed germination and seedling growth. Especially in desert grasslands with water deficiency. The aim of this study is to investigate the effects of different soil moisture contents on seed germination, seedling phenotype traits, and physiological characteristics. The following questions were raised: (1) How does soil water affect seed germination and seedling growth characteristics? (2) What are the drought adaptation mechanisms for seed germination and seedling growth of these 6 species? "
Experimental Materials and Methods
Question 12:2.1. Seed collection and storage
The section is written in sufficient detail. The methods are reproducible. However, it is notclear whether the authors developed the methods of selection and processing of seeds independently,or there are recommendations in accordance with which the following selections were made: "Duringthe collection, seeds will be directly picked from four different directions on adult plants, with 20 plants selected for each species."? Why a sample of 20 plants, and not 10 or 30? If these are standard recommendations, then this should be said and a link provided. If this methodology was developedby the authors, then this should also be stated in the text.
Response: 12: Seeds were collected by choosing 20 plants, picked in different orientations, this was to satisfy the randomness and to avoid picking a small number of seeds, which would result in inaccurate results of the experiment.
2.2. Material Handling and Collection
Question 13 The section is written in sufficient detail. The methods are reproducible. It is necessary to explain why these particular soil moisture levels were taken: "The calculated soil moisture content is 5%, 10%, 15%, 20%, and 25%...". If these are standard recommendations, then this should be stated and a link provided. If these water concentrations were chosen independently, then this choice should be explained.
Response13: This is indeed a good question, and we designed the following five soil moisture gradients with reference to the previous study of sand-fixing plant response to moisture in this region by Yang et al.
Yang, Z.; Wang, L.;Zhang, X.; et al., Seed germination and seedling growth of typical sand-fixing plants in response to soil moisture. Arid Zone Research. 2023, 41,840-842.
Question 14 2.3. Determination of seed germination parameters, seedling phenotype traits, and physiological characteristics. The section is written in sufficient detail. The methods are reproducible. It is necessary to explain the "vigor index (VI)" indicator. This indicator is not known in all countries. It is advisable to explain the source of the formula (provide a link). Is this index generally accepted, proposed by a specific scientist, or proposed by the authors? This will clarify for readers whether this index really reflects the germination energy or is it about something else?
Response14: Thank you very much for your suggestion, we have added the relevant reference.
Discussion
4.1. Seed germination response to moisture
4.2. Seedling growth response to moisture
4.3. Seedling physiological characteristics in response to moisture
4.4. Survivability of six desert shrubs in response to changes in moisture
All parts of the Discussion section contain an extended discussion of the results obtained. A detailed comparison of the results obtained by the authors with the literature data is given. There are no comments on this section.
The Conclusions section is missing!
I recommend showing in the conclusions that the goal that the authors set when planning the experiment has been achieved. And more precisely indicate how the results will be used in the future - for scientific and industrial purposes. Can the obtained results be applied in other regions of the country and the world with similar soil and climate conditions?
Response15:We have added the conclusion . “We tested the response of seed germination and seedling growth of six shrubs to moisture in a greenhouse and found that the optimum moisture for seed germination was 15-20%. In addition, we found that 15-20% was also the most favourable for seedling growth by observing the phenotypic traits and physiological characteristics of shrub seedlings. More importantly, we comprehensively analysed the seed germination parameters, seedling growth parameters, and physiological parameters of six shrubs using RDA, and found that the order of survivability was as follows: Caragana korshinskii > Zygophyllum xanthoxylum > Calligonum mongolicum > Corethrodendron scoparium > Corethrodendron fruticosu > Nitraria sibirica. These results are informative for the selection and breeding of shrubs in similar desert areas. However, our study only focused on moisture, and in later studies, we will focus on the effects of other environmental factors (e.g., temperature and salinity) [37] and plant-animal interactions [62] on the seedling growth of seed germination of these shrubs.”
Round 2
Reviewer 1 Report
Comments and Suggestions for Authors
Dear respected Authors,
Thank you for your responses to all questions.
Comments on the Quality of English LanguageEnglish writing could be improved.
Reviewer 3 Report
Comments and Suggestions for Authors
Dear colleagues!
I believe that all my comments and recommendations have been taken into account. I recommend accepting the article for publication.